# Exposure to the Death of Others during the COVID-19 Pandemic: Growing Mistrust in Medical Institutions as a Result of Personal Loss

**DOI:** 10.3390/bs13120999

**Published:** 2023-12-07

**Authors:** Brian J. Gully, Hayley Treloar Padovano, Samantha E. Clark, Gabriel J. Muro, Mollie A. Monnig

**Affiliations:** 1Department of Behavioral and Social Sciences, Center for Alcohol and Addiction Studies, Brown University, Providence, RI 02903, USA; hayley_treloar@brown.edu (H.T.P.); samantha_clark1@brown.edu (S.E.C.); gabriel_muro@brown.edu (G.J.M.); mollie_monnig@brown.edu (M.A.M.); 2Center for Addiction and Disease Risk Exacerbation, Brown University, Providence, RI 02903, USA; 3Department of Psychiatry and Human Behavior, Center for Alcohol and Addiction Studies, Brown University, Providence, RI 02903, USA

**Keywords:** COVID-19, pandemic, terror management theory, mortality salience, personal loss, death exposure, medical mistrust, public health

## Abstract

Background and aims: The prominence of death during the COVID-19 pandemic was heightened by the potential of personally knowing someone who lost their life to the virus. The terror management theory (TMT) suggests that the salient presence of death has a pronounced effect on behavior and may result in the ossification of beliefs and actions aligned with one’s worldview (i.e., the mortality salience hypothesis). In this study, we evaluated how death exposure early in the COVID-19 pandemic could enact the process of firming up held beliefs and attitudes related to health and safety. Specifically, we tested the hypothesis that exposure to a personal loss during the pandemic would strengthen participants’ baseline attitudes and behaviors regarding COVID-19 safety guidelines. Method: Data were analyzed from a prospective, regional survey administered at two time points during the pandemic, June–July 2020 and May 2021, in five United States northeastern states. Baseline and follow-up surveys were administered approximately 12 months apart, with adherence to public guidance and death exposure measured at both timepoints and other safety measures at follow-up only. Findings: Our results indicated that there were significant main effects of death exposure on guideline adherence and support for COVID-related public policy. Contrary to the mortality salience hypothesis, death exposures after baseline were related to higher medical mistrust at follow-up for those high in adherence at baseline, rather than those with low adherence. Conclusion: Our results offer some conflicting evidence to the mortality salience hypothesis. Rather than entrench people in their worldviews, death in the context of the COVID-19 pandemic appeared to sway people away from their initial stances. This finding has important implications for TMT literature and for the COVID-19 pandemic response.

## 1. Introduction 

Microscopic features of the COVID-19 virus enabled it to spread at alarming rates among human hosts, leading to a global crisis in a matter of months [1,2]. By March of 2020, the World Health organization declared COVID-19 a pandemic. The unprecedented nature of the virus prompted sweeping shelter-in-place quarantine orders and economic shutdowns of all non-essential services. In the United States, efforts to slow the spread of COVID-19 were led by the Centers for Disease Control and Prevention (CDC). Initial CDC guidelines spanned multiple anti-viral campaigns including masking, physical distancing, surface disinfection, proper hand washing practices, and the closure of non-essential businesses and spaces where people could gather in large groups [3]. Guideline adherence was generally high but marked by some non-adherence and mistrust in medical institutions [4]. Fluctuations in trust were likely the result of inconsistent public messages, particularly about masking, which was initially discouraged to preserve short supplies for medical professionals [5].

Despite measures taken to combat the spread of infection, mortality rates remained significant, particularly for older people and individuals with comorbid health conditions. By the end of 2020, COVID-19 was a top leading cause of death in the United States for individuals above the age of 35 [6]. The cascading psychological toll of the pandemic also proved insidious to younger age groups, wherein mortality as a result of suicide and drug overdoses increased. Social isolation engendered loneliness, depression, anxiety, PTSD, and insomnia [7]. Collectively, these risk factors placed an immense strain on the physical and mental health of the general public. Politicization grew and was reflected by drastic differences in the perceived severity of the virus, shifting trust in medical institutions, and acceptance/hesitancy about the COVID-19 vaccine [8]. Although there were social contributors to political polarization at this time, including the 2020 United States presidential election, constant social media/mainstream news coverage, and widespread political protests, a key component for COVID-19′s psychological effects may have been its death toll [9]. 

The terror management theory (TMT) posits that human cognition is self-reflective to such a great extent that it exposes people to the knowledge of their own mortality [10]. In order to not be paralyzed by this knowledge, it is imperative to manage death anxiety with culturally constructed worldviews that enrich one’s life with a sense of meaning and self-esteem [11]. Death reminders in daily life require psychological defenses to ignore, rationalize, or try to overcome the fear of mortality. Two distinct processes that are theorized to manage death anxiety are proximal and distal defenses [12]. Proximal defenses activate when death is clearly present in focal attention. These involve behaviors that suppress death thoughts by denying vulnerability. Distal defenses are utilized when death is not in active focus but still peripherally present. These involve bolstering one’s cultural worldview in order to fortify death-denying beliefs. 

During the COVID-19 pandemic, death became hyper-salient via mass media reports of overburdened hospitals, makeshift morgues, and the threat of rationed care [13]. Proximal defenses were those that rationalized or distanced individuals from the threat of death, driving people to either stay updated and active on the latest CDC health recommendations or to seek justifications or distractions from the disturbing news to deny vulnerability. Distal defenses were those that motivated individuals to reinforce their cultural worldviews and important aspects of their lives, perhaps by engaging in political discourse, advancing their careers, or expressing themselves in passion projects that would provide a sense of symbolic immortality.

Empirical evidence for TMT elucidates the role of distal defenses in the suppression of death thoughts. A standard practice in TMT research is to measure mortality salience effects after a delaying task in order to examine these distal defenses [14]. Reminders of death generally increase overall death anxiety unless the effect is moderated by self-esteem [15,16]. Self-esteem is closely tied to cultural worldview, as reminders of death increase in commitment to cultural worldviews and engender more fervent defenses of said worldviews when they are faced with criticism [17]. 

Support for the mortality salience hypothesis has emerged in over 25 countries, suggesting that its effects are cross-culturally applicable [18]. There are exceptions in this trend, for example, a recent study on implicit bias in a Japanese sample toward Korean surnames indicated null TMT results [19]. The researchers suggested this was a reflection of a specific cultural difference among Japanese people, but further research is needed. In drawing comparisons between Western and East Asian cultures, TMT denotes differing views on self-esteem between individualist and collectivist cultures; however, the ability of mortality salience to alter behavior by bolstering one’s commitment to a worldview has been repeatedly observed, despite individualistic or collectivist leanings [20]. 

An integration of TMT and health psychology can be found in the terror management health model (TMHM), which addresses the disparity between health- and self-oriented behaviors when confronted with death from a medical standpoint [21]. This model operates under the same framework of proximal and distal defenses. In proximal defenses, when death is salient via life-threatening health concerns, there is a motivation to promote health or, at least, remove death from one’s focal attention. In distal defenses, when death is pushed to the fringe of one’s attention, behaviors that follow are influenced by a desire to bolster esteem rather than improve health, per se. This phenomenon can be seen in scenarios where medically risky behavior, such as smoking or restricted eating, may be undertaken to maintain or bolster esteem [22,23]. The process by which the TMHM operates may relate to underlying discomfort with the fallibility of the human body [24]. When death is not an immediate threat, it is more comforting to act in accordance with a meaningful cultural worldview than it is to confront the fact that the human body is vulnerable to disease.

As discussed, some of the topics which emerged as rife with ideological division during the pandemic were vaccine acceptance/refusal and trust in medical institutions [8]. Polarization surrounding the COVID-19 pandemic in these domains may be understood via a terror management perspective, particularly through implementation of the TMHM. The TMHM perspective on the COVID-19 pandemic pays particular attention to the high prevalence of mortality throughout the pandemic and the effect this mortality salience has on behavior. Regardless of one’s views on the severity of the virus, COVID-19 is a topic highly charged with the threat of death [25]. As a result, there may be a push and pull between proximal and distal defenses during a pandemic. Proximal defenses may lead people to either promote their health or downplay the risk of disease, while distal defenses lead to the bolstering of one’s worldview in the face of global anxiety. Thus, the shift from an “in-this-together” mindset at the start of the pandemic to rapid ideological division may be explained, in part, by the methods people employ to manage death anxiety [26].

It is well established that mortality salience can lead to polarization in political domains [27]. This concept was most notable in a study which found that the personal loss of close family or friends led to reinforced political views over time [28]. Similarly, research on the death of a close other has shown that such a loss tends to lead to stronger identification with an in-group, a finding which is consistent with TMT [29]. 

In summary, the core framework of the TMHM is that in the presence of death, health behaviors and beliefs may become charged with cultural viewpoints. To explore the viability of this framework in the context of COVID-19, the present study assessed personal exposure to death during the pandemic and its subsequent effect on behavior and ideologies. We hypothesized that individuals who practiced CDC guidelines to mitigate COVID-19 transmission and were exposed to death would continue to follow CDC guidelines, become vaccinated, support COVID-19 public policy, and trust medical institutions. Conversely, we hypothesized that individuals who minimized the threat of COVID-19 and/or did not follow CDC guidelines and were then exposed to death would continue to disregard guidelines, avoid vaccination, show opposition to COVID-19 public policies, and exhibit mistrust in medical institutions. Although the latter half of this hypothesis may seem counterintuitive, it is in line with the TMHM’s perspective, particularly when one considers the substantial integration between COVID-19 attitudes and political orientation that occurred during the pandemic [30]. Overall, we expected that individuals personally exposed to death would be more polarized in their endorsement of health guidelines and public policies compared to individuals who were not personally exposed to death. 

## 2. Methods

### 2.1. Procedures and Participants 

Data were analyzed from an online survey disseminated to a sample recruited on Amazon’s Mechanical Turk (Mturk) platform by the Center for Addiction and Disease Risk Exacerbation (CADRE) at Brown University [31]. This study focused on the five U.S. states that had the highest COVID-19 deaths per capita at the time of study: New York, New Jersey, Rhode Island, Massachusetts, and Connecticut. This study was reviewed by the Brown University Institutional Review Board and determined exempt from requiring approval as a minimal risk study. The baseline survey was administered from June to July 2020. The follow-up survey was administered in May 2021. 

The participants for the present analysis were those who completed the baseline survey and the follow-up survey regarding COVID-19 safety measures (*n* = 360). To maintain the temporal sequence of death exposure preceding the outcomes, the participants who reported exposure to death at both baseline and follow-up (*n* = 75) were excluded, as the timeline of their exposure could not be isolated with available data. In addition, one participant was excluded for missing information on the racial identity covariate. The final evaluable dataset included 284 participants. 

### 2.2. Measures

Exposure to death was operationalized and coded with the question “Do you personally know anyone who has died of the novel coronavirus, or COVID-19?” with three potential answers: “Yes”, “No”, or “Possibly, but it was unclear if the cause of death was coronavirus.” For the purposes of this study, the answers “yes” and “unclear if the cause of death was coronavirus” were combined as they both involved death of a close other and fit in the TMT framework of mortality salience. Exposure to death was measured at baseline and follow-up. As noted above, the participants who reported exposure to death at baseline and follow-up were excluded due to the lack of sufficient data needed to understand the timeline of their death exposure. In order to understand the effects of death in isolation, the analysis focused on individuals who were exposed to death at follow-up only or those not exposed at all. By doing so, we ensured that exposure to death could be placed temporally between the baseline and follow-up assessments. 

Safety measures were operationalized as adherence to CDC COVID-19 guidelines, support for COVID-19 public health policy, trust in medical institutions, and vaccination status. CDC guideline adherence [31] was measured with 13 items rated on a 0–3 Likert-type scale (rarely/never, sometimes, usually, or always) with statements such as, “Remain at least 6 feet away from other people when in public?”, “Stay home as much as possible?”, and “Use a cloth face cover over your nose and mouth when in public?”. This measure was assessed at baseline and again 12 months later at follow-up. Support for COVID-19-related public policy was measured with 6 items rated on a 1–4 Likert agreement scale (α = 0.95) with statements such as “I support the closure of non-essential businesses”, and “I support the canceling or postponing of mass gatherings” [32].

Medical mistrust was measured with 7 items rated on a 1–4 Likert agreement scale (α = 0.87) with statements such as “When health care organizations make mistakes they usually cover it up”, and “Healthcare organizations don’t always keep your information totally private” [33]. 

Vaccination status was measured in the follow-up survey with the single yes/no question, “Have you received at least one dose of the COVID vaccine?”. Measures on COVID-19 public policy, medical mistrust, and vaccine status were administered at the follow-up survey only.

### 2.3. Analysis

All analyses were pre-registered (see https://osf.io/k7ayn/). Adjusted multiple regression models were implemented with covariates chosen a priori based on prior literature [34]. The demographic covariates included age, race, ethnicity, gender, SES, and perceived discrimination. Given that a high percentage of survey respondents identified as Caucasian, race was re-coded into the dichotomous variables minoritized identity and non-minoritized identity. The model-building progressed with the inclusion of these covariates (Step 1); the addition of the focal predictor, i.e., exposure to a COVID-19 death between baseline and follow-up (Step 2); the addition of the putative moderator, i.e., baseline CDC guideline adherence, (Step 3); and, finally, the inclusion of an interactive effect of recent death exposure with baseline guideline adherence (Step 4). Linear multiple regression was implemented for medical mistrust, COVID-19 public policy support, and CDC guideline adherence. In the case of the primary outcome, CDC guideline adherence, the inclusion of baseline adherence provided the means to evaluate changes in views related to safety measures against COVID-19 between baseline and follow-up. For CDC guideline adherence, the statistical power to detect interactive effects was enhanced though maintaining the original continuous nature of the variable during analysis. The published power tables indicated that the present study was sufficiently powered to detect interactive effects [35]. For vaccine status, the logistic multiple regression model examined the odds of receiving a vaccine based on baseline CDC guideline adherence and recent death exposure. The continuous variables were centered at the sample mean prior to exploring the interactions.

## 3. Results

The final sample (*n* = 284) was 53.2% female and predominantly white (74.6%) and non-Hispanic (88%). The average age of the respondents was 43 (SD = 14.5). Regarding exposure to death, 226 (79.6%) were not personally exposed to death at all, while 58 (20.4%) reported exposures at follow-up. The average CDC guideline adherence at baseline was 2.24 (SD = 0.50) and 2.05 (SD = 0.58) at follow-up. Assessed only at follow-up, support for COVID-19-related public policy was 2.91 (SD = 0.81), on average, and reported mistrust in medical institutions was 2.52, on average (SD = 0.57). CDC adherence was generally high, with no respondents indicating complete non-adherence. However, only a small portion of participants reported complete adherence to CDC guidelines, with *n* = 22 (7.75%). 

At the time of follow-up, 72.9% of the sample had received at least one dose of the COVID-19 vaccine. A significant main effect was found for death on vaccination status, such that exposure to death was predictive of having received at least one dose of the vaccine, with *b* = 0.92 (SE = 0.41), *p* = 0.024, OR = 2.52, and 95% Wald OR CI (1.13, 5.63). There was no significant interaction between CDC adherence at baseline and subsequent exposure to death on vaccination status.

The statistical assumptions were met for the covariates in the regression model, and all tolerance values were >0.88 and all variance inflation factors were < 1.13. The regression coefficients for the covariates per each outcome variable can be found in Table 1. For CDC guideline adherence, a significant main effect was found between exposure to death and adherence at follow-up, with *b* = 0.17 (SE = 0.08) and *p* = 0.048, such that exposure to death increased follow-up adherence. No significant interaction effect was found between death exposure and baseline adherence. In regard to support for COVID-19-related public policy, a significant main effect again was found for death exposure, with *b* = 0.24, (SE = 0.12) and *p* = 0.046, such that exposure to death increased support for COVID-19 public policy. No significant interaction effect was found for these variables in regard to public policy support.

In regard to medical mistrust, the relationship between death exposure and mistrust at follow-up was not significant; however, a significant main effect was found for baseline guideline adherence, with stronger adherence generally predicting less mistrust. A positive significant interaction was found between adherence and death exposure on medical mistrust, with *b* = 0.46 and *p* = 0.002. To visualize the interaction, the participants were classified as either being strongly adherent or weakly adherent to CDC guidelines based on the values above or below the centered sample mean, respectively. For the participants who were classified as being weakly adherent to guidelines, medical mistrust was higher for those without a personal death exposure (*M* = 2.63 and N = 111) than those with a death exposure (*M* = 2.51 and N = 24). Conversely, for those classified as being strongly adherent to CDC guidelines, medical mistrust was higher for those with a death exposure (*M* = 2.62 and N = 34) than those without (*M* = 2.39 and N = 115). This visualization can be found in Figure 1. The adjusted R² value for medical mistrust was 0.09 without the interaction, with F (7275) = 5.05, *p* < 0.001, and 0.14 with the interaction and F (9273) = 6.26 and *p* < 0.001, suggesting a 64% increase in the variance explained for medical mistrust when the interactive effect was included. A test of the cumulative unadjusted R² upon addition of the interaction term indicated significant improvement in the variance explained, with F = 10.19 and *p* = 0.002.

## 4. Discussion 

Based on previous research in the field of TMT, particularly, health research in the TMHM, the individuals who were highly adherent to CDC guidelines at baseline and exposed to death were expected to be more likely to be vaccinated, more trusting of medical institutions, and in favor of COVID-19-related public policy at follow-up. The individuals who were non-adherent to guidelines at baseline and exposed to death were expected to be more likely to be unvaccinated, to be against covid public policy, and to report high levels of medical mistrust at follow-up. The actual results were mixed and, in some cases, in opposition to segments of our hypotheses. The significant interactions and main effects are discussed below in their order of relatedness to the TMT hypothesis of this study. 

### 4.1. Interaction Effect of Death Exposure and CDC Adherence on Medical Mistrust

Our results offered some conflicting evidence to the mortality salience hypothesis. Rather than entrench people in their worldviews, death in the context of the COVID-19 pandemic appeared to sway people away from the initial stances they reported at baseline. Those who reported weaker adherence to CDC guidelines at baseline were subsequently more trusting of medical institutions if they were exposed to a death between baseline and follow up, whereas those who reported strong adherence to guidelines at baseline appeared to become more doubtful of medical institutions if they were exposed to death between baseline and follow up. This finding is somewhat surprising, given that most Americans viewed COVID-19 as a partisan matter poised for political polarization [26]. Consequently, it was expected that variables such as trust in medical institutions would be representative of politicized aspects of the pandemic and, therefore, a worldview that would be defended under a mortality salience threat. 

The contrary findings may be explained in several ways. One explanation may be a reflection of the complexity of death’s impact during the worst pandemic in a century. Studies reviewing the TMHM for pandemics have denoted that COVID-19 illustrated the significance of psychological reactance, defined as the perception of threats toward freedom of behavior [36]. Reactance as a variable does not appear to explain the current findings, as there was an apparent turning point with regard to medical mistrust for those exposed to death. Further research is needed to identify the potential confounding variable that may have altered the mortality salience effects and made it so that death created a re-evaluation, rather than a worldview defense. A potential variable worthy of exploration in TMT literature is general institutional trust, which has also been found to create interpersonal trust when bolstered [37]. While death may tend to entrench people in their worldviews, its effect on trust may be entirely different. It may be hypothesized that when death is hyper-present, such as when confronting personal loss, institutional trust becomes highly malleable. The implications for broader public health literature are withstanding, as the current findings highlighted the ways in which death may undercut or bolster trust in medical institutions.

### 4.2. Main Effects of Death Exposure on Safety Measures

Our analyses revealed several significant main effects of death exposure. Exposure to death positively predicted vaccination status, adherence to CDC guidelines, and support for COVID-19 public policy. Individuals who experienced the death of a close other were more likely to have received at least one dose of the vaccine, increased adherence to CDC guidelines, and increased support for COVID-19 public policy at follow up. These findings suggested that public health education efforts may benefit from devising ways to personalize the public health toll of a pandemic. 

### 4.3. Limitations

A significant limitation of the present study is the lack of attitudinal range within our sample. This sample could be described as generally adherent to COVID-19 safety measures. Overall adherence was high, and there were no instances in this sample where an individual reported total non-adherence to CDC guidelines or complete opposition to public policies. Part of this may be a reflection of the sample’s geographical make-up. New York, New Jersey, Rhode Island, Massachusetts, and Connecticut were surveyed, as they were the states with the highest death tolls during this period of the pandemic. Currently, these states also tout an over 90% vaccination rate of at least one dose, and they have among the highest rates of bivalent boosters in the country [38]. Consequently, limitations in the attitudinal range of our sample may have contributed to the insignificant interactions found between baseline CDC adherence and exposure to death on safety measures at follow-up. A nationally reflective sample, encompassing both strong adherence and strong non-adherence, may reveal alternative mortality salience effects. Further demographic limitations lie in our sample’s homogenous racial and ethnic makeup. 

Another limitation is this study’s non-traditional structure for a TMT study. Despite the measures taken to ensure the temporal order for exposure to death in the sample, this study was a post hoc analysis of cross-sectional data and not a mortality salience manipulation. Death exposure was only able to be broadly categorized, and additional information about the factors associated with the deaths, such as suddenness, was not available. Our sample was not counterbalanced, as it would be in a classical experiment, causing there to be fewer participants in the death exposure group than the no-exposure group. While this is a limitation, it is also reflective of the naturalistic sample surveyed during the pandemic. Finally, public policy support and medical mistrust were not measured at baseline, and therefore, they could not be directly compared between the time points. Future research would benefit from directly examining these variables longitudinally.

These caveats are noted not to discount findings but rather to ground findings within the scope that is being widened in the TMT field as a result of emerging discourse on replicability [39]. A recent attempt to replicate the classic mortality salience effect failed to support the classic worldview defense hypothesis, causing some researchers to call nearly 40 years of empirical support into question as a potential false positive [40]. This conclusion proved to be hyperbolic, as a reanalysis of the same dataset accounting for sufficient sample sizes indicated support for the mortality salience worldview defense hypothesis [41]. This scrutiny prompted large scale meta-analyses to encourage increased power. A preregistered *p*-curve analysis of 818 studies highlighted the importance of increased sample sizes and delaying tasks between independent and dependent variables in laboratory settings [42]. A later failure to replicate TMT effects offered more robust reasoning, suggesting that the framework for TMT research needs to be updated to not be constrained by so-called classic findings in order to be integrated into the theory [43]. This is particularly important in light of the emerging evidence that salient social norms may moderate TMT effects [44]. In other words, the ways in which death permeates the current climate, and the shifting ways culture attempts to manage it, are essential to take into account when studying TMT. The present study succeeded in the latter half of this recommendation by examining socially prevalent beliefs which are under scrutiny in the zeitgeist. Its contrary-to-predicted findings may offer an expansion on understanding of the ways death exposure exhibits influence.

## 5. Conclusions

While death was hyper-prevalent within discourse during the COVID-19 pandemic, its real-world reach was far more personally significant for those who directly experienced loss. The results of this study suggested that personal exposure to death increased several safety-related behaviors. Personal loss also predicted increased medical trust at follow-up for those who began with low adherence to CDC guidelines. However, death exposure for those who were strongly adherent to guidelines served to erode trust in medical institutions. These findings are yet to be fully understood, and the variable of trust in relation to major institutions is one that required attention, but the results undoubtedly hold significant implications for TMT literature, broader pandemic literature, and the field of public health as a whole.

## Figures and Tables

**Figure 1 behavsci-13-00999-f001:**
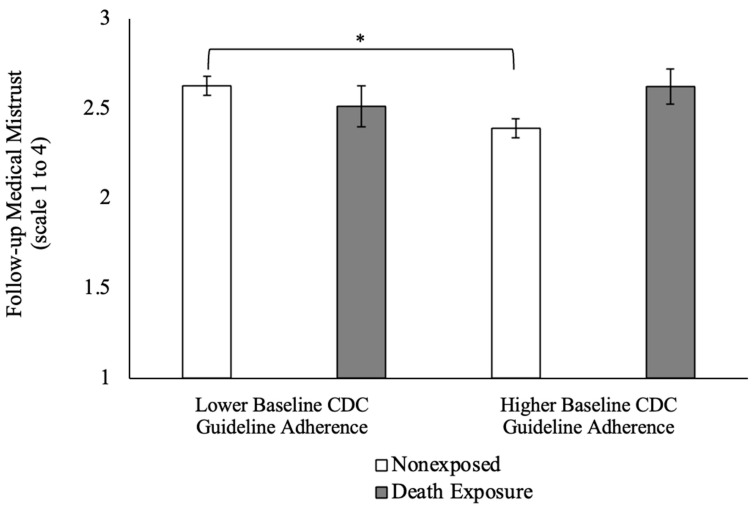
Interaction between death exposure and CDC guideline adherence level on medical mistrust. *, *p* < 0.05.

**Table 1 behavsci-13-00999-t001:** Estimates (and standard errors) from the regression models evaluating the effects of the covariates and death exposure on the outcome variables. *, *p* < 0.05 and **, *p* < 0.001.

	Covid Public Policy	Vaccination Status(Logistic)	CDC Guideline Adherence	Medical Mistrust
	*b* (SE)	*b* (SE)	*b* (SE)	*b* (SE)
Age	−0.00 (0.00)	−0.01 (0.01)	−0.00 (.000)	−0.00 (0.00)
Gender	0.09 (0.10)	−0.32 (0.29)	0.22 (0.07) *	−0.02 (0.06)
Race	0.13 (0.11)	−0.61 (0.33)	0.10 (0.08)	0.18 (0.08)
Ethnicity	−0.26 (0.15)	−0.71 (0.43)	−0.03 (0.11)	0.05 (0.10)
Socioeconomic status	−0.06 (0.03) *	0.31 (0.08) **	−0.02 (0.02)	−0.06 (0.02) *
Perceived discrimination	−0.09 (0.05)	−0.14 (0.14)	−0.06 (0.04)	0.98 (0.03) *
Death exposure	0.24 (0.12) *	0.92 (0.41) *	0.17 (0.08) *	0.03 (0.08)
F-value (7, 275)/chi-square	2.66 *	30.41 **	3.25 *	5.05 **
Variance explained (R²)	0.04	0.15	0.05	0.09

Note: For the logistic regression model, the likelihood ratio, Chi-Square test, and Nagelkerke pseudo-R² values are reported. For all other models, the model-adjusted F and R² values are reported.

## Data Availability

The datasets analyzed during the current study are available from the corresponding author on reasonable request.

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
