# Peer review of "Exposure to the Death of Others during the COVID-19 Pandemic: Growing Mistrust in Medical Institutions as a Result of Personal Loss"

_behavsci, 2023, doi:10.3390/bs13120999_

Round 1
Reviewer 1 Report
Comments and Suggestions for Authors
The present work addressed a critical issue in healthcare scenarios and their actors in the context of death anxiety in the COVID-19 pandemic in USA: their adherence and support to CDC (Centers for Disease Control and Prevention) guidelines, as well trust in medical institutions. Despite these behaviors were found to be generally high, some polarization was also observed (non-adherence and mistrust). The authors provide evidence of the contribution of Terror Management Theory (TMT) in medical scenarios (TMHM) and work under the hypothesis that individuals personally exposed to death would be more polarized in their endorsement of guidelines than those not personally exposed.The conceptual frameworks are well described and presented, thus providing a good material for those unfamiliar with it. The main limitations are identified and discussed, and some others (see below) could be added.
The results yield new data on the role of TMT in this scenario, and the confrontation with other hypotheses (in particular, the mortality salience hypothesis) should be foreseen due to the complexity of human scenarios and behaviors that went into the limits in these severe worldwide. Also, as the authors discuss to the still emerging knowledge on TMT The results also open new questions that need further study and understanding.
The sample (n = 284) was quite homogeneous, and is referred as “consisted of a majority female (53.2%), white (74.6%), and non-Hispanic (88%)”.
1) The percentage 53.2% for female sex/gender is referred as ‘majority’, something that it is true in politics but no were else. Please, reconsider the sentences, as it is true for the other sociodemographic variables, but not this, as it means 133 participants were males (46.8%).
2) In this last respect, can results per sex/gender be presented and discussed?
3) Also, despite the authors already referring to culture in the introduction, can they provide some further discussion/expectations with other ethnic groups?
The results would benefit from a figure representation or the whole work as a graphical abstract.
Please, change the word ‘elderly’ (considered ageist) to ‘old people’ or ‘older people’. Thanks
Reviewer 2 Report
Comments and Suggestions for Authors
The submitted manuscript addresses important and potentially resulting in long-lasting effect phenomenon. However, before suggesting it for publication, I would like to focus on several issues:
#1. In the abstract you are writing about northeastern states, but did not precise of which country.
#2. The introduction is lengthy and vaguely in terms of supporting the hypotheses. The Authors should focus on the main background theory (TMT) and the role of an exposure to death and skip the general introduction about pandemic (e.g., make 1-2 sentences of summary regarding the prevalence of death during the pandemic).
#3. How the sample size was determined form the study? How was determined the part of the participants who were exposed to death?
#4. Please describe the statistical power or sensitivity in the study to detect interaction. When referring to means in interaction, please, note how many participants was in the group (e.g., low adherence and exposure to death; these group seem to be very small).
#5. Some tables or figures need to be added to clarify the results. Which were the fit statistics for regression models? Which were the correlations between predictor variables and dependent variables?
#6. Discussion needs some introductory paragraph summarizing the predictions. The effects of death exposure should be discussed before the interactions.
#7. Another limitation is that the Authors did not control for the factors associated with death. For example, age of person who died, whether the death was sudden or slower, etc.
Comments on the Quality of English LanguageEnglish is understandable. However, anthropomorphisations could be corrected.
Reviewer 3 Report
Comments and Suggestions for Authors
I have found this study interesting and well-grounded. Notwithstanding, my main question is about the correct derivation of the hypothesis posed by the TMHM. In the last paragraph of section 1, the hypotheses are exposed, but not their logical derivation from TMHM. Could they be elaborated more explicitly? In the study there are two hypotheses:
H1:
a) those who practiced CDC guidelines & are exposed to death à continue CDD guidelines, support public policy and trust medical institutions;
b) those who do not follow CDC guidelines & minimized the covid threat & are exposed to death à opposition to public policies and mistrust in medical institutions
H2: people exposed to death would be more polarized about endorsement to CDC than those not exposed to death (in the second moment of assessment).
How do you derive them from TMHM?
- There is a reference to “delaying tasks” In the last paragraph of page 2 that is not clear in the context of death anxiety. Can the authors explain it more thoroughly? This topic appears again in the Limitations section.
- The authors use an undefined acronym (MS) in the same paragraph. Please clarify it.
- There is no clear understanding in the paper about the exception of the Otsubo & Yamaguchi (2023) study concerning the mortality salience hypothesis.
- The results section should inform the testing of statistical assumptions for multiple regression, mainly for the covariates. In my experience, covariates are very tricky.
- The scales of the Exposure to Death, Safety Measures and Medical Mistrust are not the same scale, although they have four levels, but why was the Safety Measure scored from 0-3 instead of 1-4 as the others?
- What is the meaning of the Est. symbol in the results section? It seems to mean “Estimation”, but it is not clear its meaning (probably is the slope estimation).
- Was not a dependent measure the difference between pre and post-test in the safety measure, support for covid-19 and medical mistrust? The authors only analyzed the post-test scores, but maybe the difference between pre and post would also be interesting.
- What does mean “a negative main effect”?
- Although the authors present several limitations, there is one that I consider interesting: they measured safety measures, exposure to death and medical mistrust only on two temporal points. This is not a longitudinal online survey. A more dense assessment of these variables in time would be more fruitful to disentangle the pattern of results from exposure to death.
Minor points
- The paragraph right before methods has an incorrect line.
Round 2
Reviewer 1 Report
Comments and Suggestions for Authors
The authors have properly done the corrections and the amendements requested to fill some small gaps or to clarify issues of interest, as highlighted in several parts of the revised ms.
Author Response
Thank you for your review. We are happy to have addressed the comments you provided.
Reviewer 2 Report
Comments and Suggestions for Authors
After reading the revised version of the manuscript I have further suggestions.
#1. The introduction failed to fully justify the measurement of medical mistrust and vaccinations intentions. Please, expand your argumentation regarding these variables in the introduction.
#2. The Authors did not provide an estimation of sensitivity of the study to detect an interaction effect. Please, include power considerations in the analysis section.
#3. The regression models were presented without information about their fit to data (F, R square) - please add this information about all models. Est. should be replaced with B. Why the interaction effect is not provided in the table 2?
#4. Please, specify how many individuals were in low/high adherence x death exposure groups (Figure 1).
Comments on the Quality of English LanguageSentences such as "results showed" should be corrected to active voice like We showed that...
Round 3
Reviewer 2 Report
Comments and Suggestions for Authors
The Authors revise their manuscript according to the suggestions. I ahve minor suggestions to the upadted version.
#1. When refering to regression models R square is not enough. You have to report F for significance of the model(s) and F for change in R2 to justify the interaction term. In interaction which are SD for all groups and which post-hoc test has been used to compare the group means.
#2. Now it is visible that groups in interaction have different number of participants. I suggest to refer to these differences in the limitations section. Did the number of participants in conditions could affect the results?
#3. The interaction term added to the model for medical mistrust explain additional 5% of DV's variation. So, please use delta R square and F for change to indicate that the interaction was significant.
